# Fatigue and associated factors among adult cancer patients receiving cancer treatment at oncology unit in Amhara region, Ethiopia

**Lingerew Animaw**[1], **Teshager Woldegiorgis Abate**[2], **Destaw Endeshaw**[2], **Dejen Tsegaye** [3] *

**1** Adult Health Nursing at Feres Bet primary hospital, Amhara, Ethiopia, **2** Department of Adult Health Nursing, College of Medicine and Health Science, Bahir Dar University, Bahir Dar, Ethiopia, **3** Department of Nursing, Debre Markos University, College of Health Sciences, Debre Markos, Ethiopia

* dejenetsegaye8@gmail.com

## Abstract

### Introduction

Fatigue is one of the most commonly and frequently reported symptoms by cancer patients. The cause of fatigue is multifactorial in origin, and its impact varies in range from affecting patients' daily social life, and physical, mental, economic, and social well-being to becoming a threat to their quality of life. Therefore every cancer patient needs to be screened for fatigue and considered as one of the vital signs.

### Objective

To assess the prevalence of fatigue and associated factors among adult cancer patients, receiving cancer treatment at the oncology unit in Amhara region, Ethiopia, 2022.

### Method

Institutional-based, cross-sectional study was conducted among adult cancer patients receiving cancer treatment from May 9th–June 8th, 2022. A stratified random sampling technique was used to select study participants. Data were entered into Epi data version 4.6 and then exported to the SPSS statistical package version 23 for further analysis. Both bivariable and multivariable logistic regression analyses were carried out. P-values <0.05 in multivariable logistic regression were considered statistically significant.

### Results

The prevalence of cancer-related fatigue was 77.3% at 95% CI (73.1–81.1) with nonresponse rate of 1.97% (9). Poor social support (AOR = 3.62; 95% CI: 1.53–8.60), anxiety (AOR = 3.13; 95% CI: 1.54–6.36), physical inactivity (AOR = 3.67; 95% CI: 1.74–7.54), underweight (AOR = 2.03; 95% CI: 1.05–3.90), anemia (AOR = 2.01; 95% CI: 1.04–3.90), surgery as a treatment modality (AOR = 0.21; 95% CI: 0.06–0.78), combination therapy (AOR = 3.56; 95% CI: 1.68–7.54), treatment less than 3 cycle (AOR = 4.43; 95% CI: 1.53–

**Data Availability Statement:** All relevant data are within the paper and its Supporting Information files.

**Funding:** The authors received no specific funding for this work.

**Competing interests:** The authors have declared that no competing interests exist.

**Abbreviations:** AOR, Adjusted Odds Ratio; ASSIST, Alcohol Smoking Substance Involvements Screening Test; BFI, Brief Fatigue Inventory; CCI, Charles Comorbidity Index; CI, Confidence Interval; COR, Crude Odds Ratio; DCSH, Dessie Comprehensive Specialized Hospital; FCSH, Felegehiwot Comprehensive Specialized Hospital; FMOH, Federal Ministry of Health; GAD7, Generalized Anxiety Disorder-7; GCSH, Gonder Comprehensive Specialized Hospital; GSLTPAQ, Godin-Shephard Leisure-Time Physical Activity Questionnaire; LSI, Leisure Score Index; NCCN, National Comprehensive Cancer Network; PHQ9, Patient Health Questionnaire -9; PROMIS, Patient Reported Outcomes Measurement Information System; QOL, Quality of Life; SPSS, Statistical Product and Service Solution; TCSH, Tibebegion comprehensive specialized hospital; WHO, World Health Organization.

12.80), and treatment 3–5 cycle (AOR = 3.55; 95% CI: 1.38–9.09) were significantly associated factors with cancer related fatigue.

## Conclusion

Psychosocial assessment and intervention, nutritional support, early intervention of anemia, and promoting exercise are the key elements to minimizing fatigue among cancer patients.

## Introduction

Cancer is a large group of diseases that can start in almost any organ or tissue of the body when abnormally uncontrollable cells grow and metastasize to invade other parts of the body [1]. It could be one of the worldwide major open wellbeing issue, which influences all people and is the second leading cause of mortality after cardiovascular disease [2]. Worldwide, around 19.3 million new cancer cases and nearly 10.0 million cancer deaths happened in 2020 [2].

The most widely used definition of Fatigue among cancer patients is by the national comprehensive cancer network (NCCN) which is "a distressing persistent subjective sense of tiredness or exhaustion because of malignancy or treatment and interferes with usual functioning" [3]. It involves physical, mental, and emotional manifestations, including generalized weakness, reduced concentration or attention, demenishied motivation or interest to engage in usual activities [4].

Surgery, radiation, and chemotherapy, can induce inflammation as a result of cellular damage and tissue injury [5]. This tissue injury may cause elevation of cytokine and previous studies confirmed that circulating cytokine is correlated with fatigue [4,6]. Chemotherapy uses chemical agents to destroy cancer cells in the body and inhibit the growth and the dissemination of cancerous cells [7]. It was nonspecifically targeted skeletal musculature, especially the mitochondria, inducing adverse side effects due to low energy supply and high oxidative stress [8,9]. Once the structure and function of mitochondria are disrupted, the energy supply of cells decreases, leading to various unpleasant conditions, like fatigue, muscle wasting, reduced exercise capacity, that are due to cancer treatment [8,10]. This leads persistent worry or sense of tiredness that can happen regardless of cancer type and could be involved physically, emotionally, and cognitively, that is due to tumor or its treatment then to fatigue [11].

Fatigue is categorized either as primary or secondary [12]. Primary fatigue is usually due to the tumour itself or changes in cytokines produced during cancer treatment or as hypothalamic regulatory circuits [8]. Secondary fatigue is linked to one or more disease-related symptom(s) such as sleep disturbance, malnutrition, anaemia, and commodities [13].

There is no specific diagnostic algorithm for fatigue in the past, but now there was an effort to address what is fatigue and specific diagnostic criteria were developed [3]. More than 20 different unidimensional and multi-dimensional tools have been applied in practice [14]. Among the unidimensional tools, brief fatigue inventry (BFI) are the most widly used which focused on the identification of the occurrence and severity of fatigue [15], while multi-dimensional ones examine its impact on physical, social, emotional, and cognitive functioning [14].

The intervention begins by identifying the reversible contributing factors like unrelieved pain, emotional distress, sleep disturbance, anemia, nutritional complication, comorbidities, medication side effects, any psychosocial disturbance like depression and anxiety and decreased activity level [3]. The management of fatigue incorporates both pharmacological

and non-pharmacological approach [6,10,16]. From the non-pharmacological methods, patient education, exercise, psychosocial education, suitable nutrition and hydration, cognitive behaviour therapy, and relaxation are the best technique to alleviate fatigue [6,16–18]. Whereas on pharmacological management there is no common agreement. In clinical practice, cortico-steroids are the most widely used medication in cancer patients to alleviate inflammatory pain, relieve nausea, reduce itching, and finally reduce fatigue [17,19]. Many scholars agree on the psycho-stimulants group, which incorporates methylphenidate and modafinil [10,19,20].

The burden of cancer incidence and mortality is rapidly growing worldwide. This is due to the growth of the population as well as changes in the prevalence and distribution of the main risk factors for cancer and poor socioeconomic development [8,21]. Thus, the prevalence of patients who experience fatigue varies across studies, which range from 14% to 99% depending on the type of treatment, the type of cancer, stage of cancer, and method of assessment [22–25].

Fatigue has a wide range of consequences, including disrupting physical, mental, economic, and social well-being as well as being a risk factor for decreased survival, with a strong positive association for all cancer-related mortality [26]. Untreated fatigue may result in decreased satisfaction for both the patient and the caregiver and lead to expanded costs related to a number of phone calls, clinic visits, and critical care room visits, as well as postponed timing and dose of treatments [27], dependency on others, loss of power over decision making, and daily living disruption [28]. Fatigue causes a reduction in survival and interferes with job, enjoyment of life, relationships, and motivation due to the cancer [29].

In Ethiopia, national cancer control strategies are developed based on the WHO global cancer control strategy that incorporates cancer prevention, screening, diagnosis, treatment, and palliative care for cancer population [30,31]. However, it does not incorporate fatigue in different types of service [31] like pain; as a result, the factors associated with fatigue are not well known. This is one of the most serious challenges to effectively managing fatigue. Therefore, the primary goal of the current study is to evaluate the prevalence of fatigue and its contributing factors among adult cancer patients undergoing cancer therapy at the oncology unit of the cancer treatment center in the Amhara region.

## Methods and materials

### Study design, and period

Institutional-based cross-sectional study design was employed among adult cancer patients receiving cancer treatment at an oncology unit from May 9 to June 8, 2022.

### Study setting

The study was carried out in the cancer treatment hospitals in the Amhara region, Ethiopia. There are four hospitals in the area, each of which has a cancer treatment facility. The hospitals were Gondar Comprehensive Specialized Hospital (GCSH), Felegehiwot Comprehensive Specialized Hospital (FCSH), Tibebegion Comprehensive Specialized Hospital (TCSH), and Dessie Comprehensive Specialized Hospital (DCSH). The distances of these hospitals from Addis Ababa, the capital city of Ethiopia, are 564 km, 748 km, 399 km, and 480 km, respectively. These hospitals have an oncology clinic or treatment center with options for both inpatient and outpatient cancer diagnosis and care. For the treatment of cancer patients, the FCSH, GCSH, DCSH, and TCSH oncology units now have 18, 32, 15, and 8 beds, respectively. The service is offered by nurses, oncologists, and general practitioners.

**Populations.**   All adult ($>$ 18 years old) cancer patients in Ethiopia's Amhara region who were undergoing cancer treatment at the oncology unit were considered a source populations

and all adult cancer patients getting cancer therapy at oncology units in the Amhara region of Ethiopia during the data collecting period were the study populations. The study unit consisted of adult cancer patients who were chosen at random during the data collection period.

**Eligibility criteria.** Cancer patients receiving treatment were included in the study whereas cancer patients who were newly diagnosed at the time of data collection, critically ill, hearing, and/or intellectually impaired (unable to respond appropriately to a question) were excluded from the study.

## Sample size determination

Both the first and second objectives were utilized to calculate the sample size. According to the single population proportion assumption, the study sample size for the first objective was calculated as follows using the proportion of 74.8% [25], the 95% confidence interval, and the 5% margin of error.

$$n = \frac{(Z\,\alpha/2)^{2\,\dot{x}}\,p\,(1-p)}{(d)^2}$$

n = ≈ 290
Where: n = Minimum sample size
P = Estimated proportion of CRF (74.8%)
d = the margin of sampling error tolerated (5%)
$Z\alpha/2$ = is the standard normal distribution at 1-α% = confidence level (95% = 1.96)
n = 290 by adding 5% non-response rate, the total sample size were 305.

With regard to the second objective, the sample size was computed using the Epi info 7 software under the following assumptions: 95% confidence interval, 80% power, and a 1:1 ratio of exposed to unexposed subjects (Table 1).

**Table 1. Sample size determination based on second objective for fatigue and associated factors among adult cancer patients receiving cancer treatment at oncology unit in Amhara region, Ethiopia, 2022.**

| S.no | Reference | Variables | | Proportion in % | RR | OR | Sample size |
|---|---|---|---|---|---|---|---|
| 1 | Research conducted at Tikur Anibesa Ethiopia [25] | Age | Exposed Age (41–60) | 84.6% | 1.32 | 3.13 | 154 |
| | | | Un exposed Age (20–40) | 63.7% | | | |
| 2 | | Comorbidity | Exposed Comorbidity | 94% | 4.87 | 65.5 | 18 |
| | | | Un exposed No comorbidity | 19.3% | | | |
| 3 | | Stage of cancer | Exposed Stage III | 69.2% | 0.73 | 0.15 | 94 |
| | | | Un exposed Stage IV | 93.6% | | | |
| 4 | | Type of cancer | Exposed Breast cancer | 77.9% | 1.60 | 3.72 | 96 |
| | | | Un exposed Colorectal cancer | 48.6% | | | |
| 5 | | Type of treatment | Exposed Chemotherapy | 61.2% | 0.68 | 0.19 | 90 |
| | | | Un exposed Radiotherapy | 88.9% | | | |
| 6 | Maximum sample size | | | | | | 154 |

It was discovered that 154 was the maximum sample size determined using power approach. Taking into account a 5% non-response rate resulted in a final sample size of 162. This, however, was a smaller sample size than determined by the single population proportion formula. To increase representativeness, the maximum sample size (305) determined using the first objective was chosen. A design effect was found to be essential since several hospitals were employed, each with a varied population size and service type. As a result, the final calculated sample size (305) was multiplied by 1.5 to produce 458. Thus, 458 adult cancer patients made up the total sample size for this study.

## Sampling procedure

The participants in the study were chosen using a stratified sampling technique. In the Amhara regional state, there are four cancer treatment hospitals: FCSH, TCSH, GCSH, and DCSH. According to respective sample sizes, cancer patients were allocated proportionally. By dividing the anticipated total number of cancer patients expected to be receiving active follow-up by the desired sample size for each institution, the sampling interval (Kth patient) was calculated. Until the necessary sample size was attained during the study period, study subjects were recruited for every $K^{th}$ patient (Fig 1).

$ni = \frac{n \times Ni}{N}$ Where:

$ni$ = is the sample size of the $i^{th}$ hospital required

$Ni$ = is population size of the $i^{th}$ hospitals from medical records

$n = n1+ n2+n3$ is the total sample size (458)

$N$ = is the total population size of all the three hospital

**Dependent variable.** Fatigue among cancer patients.

**Independent variables.** Socio-demographic characteristics: Sex, age, marital status, have children, educational status, income, residence, occupational status, and medical payment, Clinical characteristics of the disease and treatment variables: Treatment type, cancer type, stage, cycle of treatment, anemia, pain, antiemetic, body mass index, pain treatment modalities, and comorbidities, and Psychological and behavioral variables: Depressions, anxiety, sleep disturbance, smoking status, alcohol drinking, social support, and exercise.

## Operational definition

The operational definitions of different variables are summarized in a table (Table 2).

## Data collection tools and personnel

The data were collected by using validated interviewer-administered questionnaire [37,40,41,43–47]. Face-to-face interviews, which last 30 minutes for each patient, were used to complete the questionnaire. The questionnaire had nine parts; the first part would contain patient socio-demographic information. The second part contains a validated nine-item numeric rating BFI scale questionnaire to measure CRF, originated from [45] and validated in Ethiopia with the overall Cronbach's alpha of 0.97 [15]. The first three items represent the severity of fatigue at its worst, usual fatigue, and fatigue now, with 0 representing no fatigue and 10 representing the worst fatigue you can imagine. The remaining six items assess the interference of fatigue with general activity, mood, walking ability, normal work, relations with other people, and enjoyment of life in the past 24 hours. The interference items range from 0 being doesn't interfere to 10 being completely interfering. The mean score could be interpreted as 0 for no fatigue, 1–3 for mild fatigue, 4–6 for moderate fatigue, and 7 or more for severe fatigue [15,45].

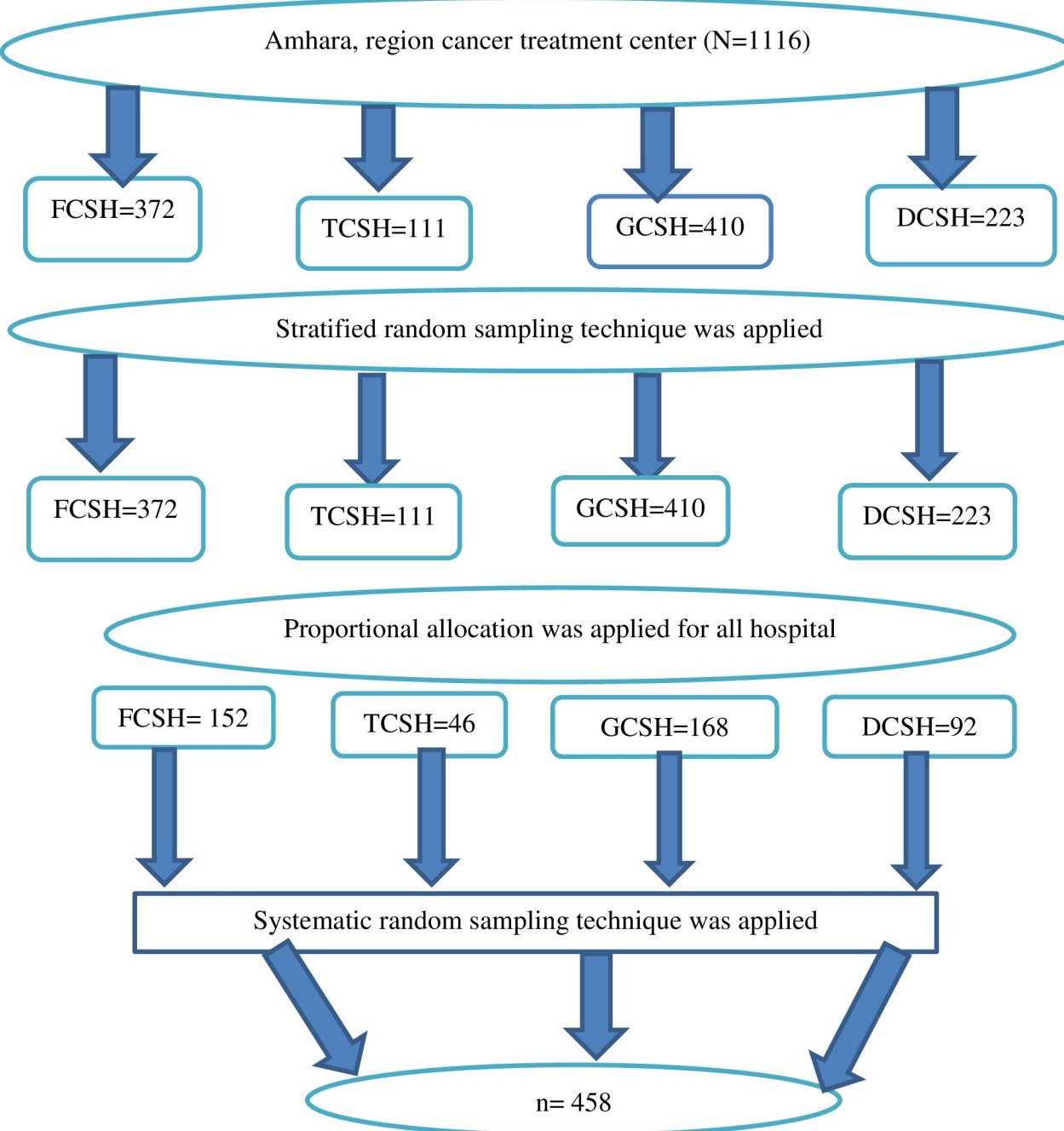

**Fig 1. Schematic representation of sampling procedure for fatigue and associated factors among adult cancer patients receiving cancer treatment at oncology unit in Amhara region, Ethiopia, 2022.**

The third part was contain the PHQ 9 [46] and GAD7 [37] questionnaire to measure anxiety and depression. The PHQ9 is a 9-item questionnaire with a response rate of not at all (0) to nearly every day (3) and an overall sum range from 0 to 27. Scoring can be done by summing up the overall response and the higher score, which is indicative of the higher level of depression. A score of greater than 10 is clinically meaningful depression. GAD-7 has a 7-item questionnaire used to measure anxiety disorders with a response rate of not at all (0) to nearly

**Table 2. Operational definition for fatigue and associated factors among adult cancer patients receiving cancer treatment at oncology unit in Amhara region, Ethiopia, 2022.**

| No | Variable | Operational definition |
|---|---|---|
| 1 | Fatigue | When patients respond based on BFI sub-scale and as a mean BFI score of ≥4 [3,17,25,32]. |
| 2 | Physically active | When patients report physical activities based on the Godin-Shephard Leisure-Time Physical Activity Questionnaire (GSLTPAQ), and the leisure score index (LSI) ≥ 24 are considered physically active whereas individuals with LSI ≤ 23 are classified as physically inactive [33]. |
| 3 | Depression | When the patients respond based on the patient health question (PHQ), and if the overall scores are ≥10 [34–36]. |
| 4 | Anxiety | When the patients respond based on the Generalized Anxiety Disorder (GAD) questionnaires, and if the overall scores are ≥10 [37,38]. |
| 5 | Anemia | When the patient's hemoglobin level is less than 13 mg/dl for men and less than 12 mg/dl for women, which is recommended by WHO [39]. |
| 6 | Sleep disturbance | When the patients respond based on the PROMIS (patient reported outcomes' measurement information system) sleep disturbance sub-scale, and the overall row score ≤ 25 as no sleep disturbance, 26–29 as mild, 30–37 as moderate, and ≥ 38 as severe [40]. |
| 7 | Smoking | When the patients respond based on the ASSIST (alcohol, smoking, substance, involvement screening test) scale and the sum of the questions is from 0–3 to consider it as low risk, 4–26 moderate risk, and ≥27 as high risk [41]. |
| 8 | Alcohol | When the patients respond based on the ASSIST scale and the overall sum of the questioners is from 0–10 as low risk, 11–26 as moderate risk, and ≥27 as high risk [41]. |
| 9 | Social support | When the patients respond based on the Oslo social support scale, overall sum from 3 to 8 is poor social support, 9–11 moderate social support and 12 to 14 strong social support [42]. |
| 10 | Comorbidities | Score is based on the Charles comorbidity index (CCI), and when the scores is 1–2 as mild, 3–4 as moderate, and ≥5 as severe [43]. |

every day (3) and the overall score ranges between 0–21. A score of greater than 10 is clinically meaningful anxiety.

The fourth part of the questioner's was sleep assessment by PROMIS sleep disturbance scale with an eight-item scale [40]. Each item is rated on a 5-point Likert scale, with a range in score from 8 to 40. The raw scores on the eight items should be summed up to obtain a total raw score. A score of less than 25 indicates no sleep disturbance, 26–29 indicates mild, 30–37 indicates moderate, and 38 or more indicates severe.

The fifth part was an exercise questionnaire derived from the four-item GSLTPAQ to assess the physical activity of the patients [33]. Scores are calculated based on weekly LSI, in which the number of physical activities for each item is multiplied by 3 for mild activities, by 5 for moderate activities, and by 9 for strenuous activities, and then summed up. The scores are based on the North American public health physical activity guidelines, as individuals reporting LSI ≥ 24 are classified as active, whereas individuals reporting LSI ≤ 23 are classified as insufficiently active [33].

The sixth section consisted of eight-item alcohol and smoking status questioner adapted from the WHO ASSIST tool [41]. The score is interpreted as the sum of all items except question number one. If the summed part for alcohol ranges from 0–10 as low, from 11–26 as moderate, and if ≥27 as high, for tobacco smoke, it is interpreted as 0–3 low, 4–26 moderate, and ≥ 27 as high risk [41].

The seventh part of the questionnaire was the social support questioner from Oslo Social Support Questioner [42]. The overall sum score ranges from 3 to 14, and interpretation is categorized into three broad categories of social support, as 3–8 for poor social support, as 9–11 for moderate social support, and as 12–14 for strong social support [42].

The eighth part of the questionnaire was assessments of the comorbidities, which are derived from the CCI [47]. It has 19 conditions. Of these, 3 are stratified according to severity,

which are weighted differently based on their mortality association and then added up to give the index score, which varies from 0 to 33. But since this study is on cancer patients, cancer-related diagnoses are removed from the CCI. Interpretation depends on the severity of comorbidity and is categorized into three grades: mild, with CCI scores of 1–2; moderate, with CCI scores of 3–4; and severe, with CCI scores ≥5 [43].

The final part was a checklist to retrieve some important information from the chart, like stage of cancer, type of cancer, type of treatment, pain treatment modality etc. which was not accessed by patient interviews.

### Data quality control

To assure the quality of the data, the following measures were undertaken. The questionnaire was adopted from the previous study [15]. In addition, training was given for data collectors about objectives, questionnaires, eligibility criteria, and data collection processes supervised by three MSC supervisors, and the principal investigator was receiving the report by checking its completeness daily.

### Data management and analysis

First data were coded; completeness and consistencies of questionnaires were checked every day. Double data entry was made using Epi data version 3.1 software and then the data were exported to SPSS statistical package version 25 for further analysis. Before analysis, data were cleaned for possible errors. Bivariate and multivariate analysis was carried out to identify variables that are significantly associated with CRF. Hosmer and Lemeshow test were performed to test for model fitness and was 0.287. Variables whose p value less than 0.25 in bivariate analysis and which can fit for model of regression were included in multivariable logistic regression. AOR at 95% CI with p-value < 0.05 was estimated to identify the associated factors on multivariable logistic regression. Variables was interpreted as having statistically significant association when $p \leq 0.05$ in logistic regression, then the results were presented in the form of figures, tables and graphs.

### Ethical consideration

Ethical clearance was obtained from Bahirdar University College of Medicine and Health Science College Ethical Review Committee (Reference number HSC/1423/19/22) then the official letter was submitted to FCSH, GCSH, TCSH, and DCSH, to get permission from respective directors. Written consent was obtained from all participants. Also affirmation that they are free to withdraw consent and to discontinue participation was made. Privacy and confidentiality of collected information was ensured throughout the process as no name is written. Privacy and confidentiality of collected information was ensured throughout the process as no name is written. This study was conducted in accordance with the declaration of Helsinki.

## Results

### Socio-demographic characteristic of the respondents

Four hundred forty nine of the patients from the entire sample size took part, equating to a response rate of 98.03%. Of them, 239 (55.9%) were married, and 295 (65.7%) were females. The patients' median age was 47, with a range of 18 to 80 years. Likewise, the patients' median monthly income was 3000 birr (Fig 2). More than three forth 361(80.4%) of the patients have no children and nearly two-thirds 296(65.9%) of the patients are residing in rural areas (Table 3).

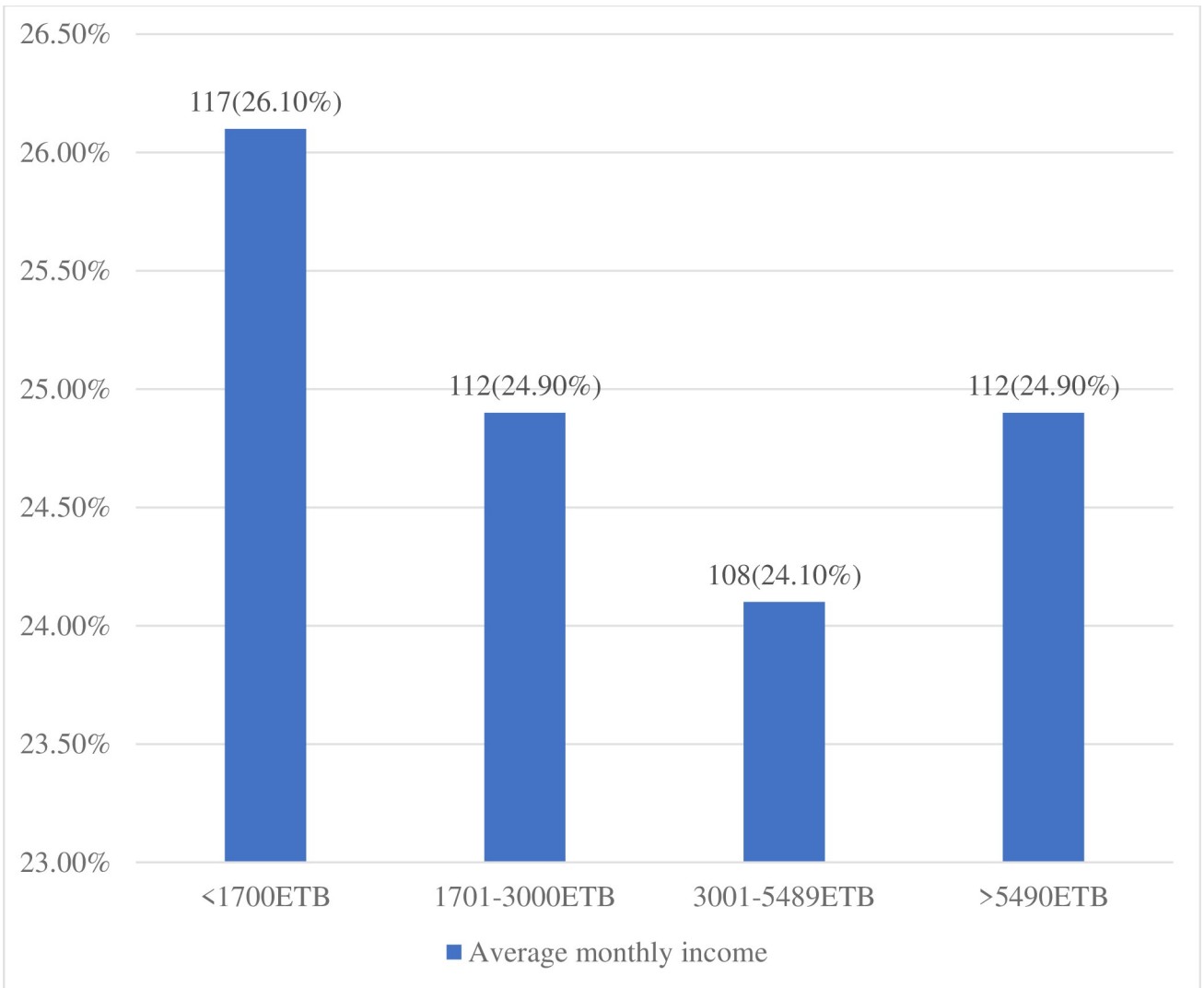

**Fig 2. Average monthly income of adult cancer patients receiving cancer treatment at oncology unit in Amhara region, Ethiopia, 2022 (N = 449).**

### Psychosocial and behavioral characteristics of the participants

More over half of the total number of patients (239, or 53.2%) were physically active. Nearly three-fourths of 336 patients (74.8%) and the majority of 399 patients (88.9%) were at lower risk groups for alcohol consumption and tobacco smoking respectively. According to the Oslo social support scale, 213 (47.4%) patients had low social support, while 277 (61.7%) patients felt anxiety (Table 4).

### Clinical characteristics of the participants

Out of the 449 cancer patients who participated in the study, 246 (54.8%) were anemic, 362 (80.6%) used antiemetic drugs to prevent nausea and vomiting while undergoing cancer treatment, 92 (20.5%) experienced severe pain, only 43 (18%) used moderate and strong opioids, and 272 (60.6%) had advanced stages of the disease. Only 24 (5.3%) of the patients had five or more comorbidities, as measured by the CCI Table 5. Gastrointestinal & genitourinary system cancer ranked first 125(27.8%), followed by breast cancer 99(22%) (Fig 3).

**Table 3. Sociodemographic characteristics of adult cancer patients receiving cancer treatment at oncology unit in Amhara region, Ethiopia, 2022 (N = 449).**

| Variables | Categories | Frequency | 100% |
|---|---|---|---|
| Age | 18–40 | 142 | 31.6% |
| | 41–60 | 228 | 50.8% |
| | >60 | 79 | 17.6% |
| Marital status | Single | 81 | 18.0% |
| | Married | 239 | 55.9% |
| | Divorced | 70 | 13.8% |
| | Widowed | 59 | 12.2% |
| Education status | Unable to read & write | 212 | 47.2% |
| | Able to read & write | 86 | 19.2% |
| | Primary | 52 | 11.6% |
| | Secondary | 45 | 6.7% |
| | Collage &above | 54 | 12% |
| Occupation | Civil servant | 57 | 12.7% |
| | Private | 29 | 6.5% |
| | Merchant | 62 | 13.8% |
| | Farmer | 253 | 56.3% |
| | Students & house wife | 48 | 8.2% |
| Medical payment | Self | 94 | 20.9% |
| | Employee | 50 | 11.1% |
| | CBHI | 305 | 67.9% |

## Prevalence of fatigue among adult cancer patient receiving cancer treatment

With a mean score of 5.3 ± 2.4SD, adult cancer patients had a prevalence of CRF of 77.3% (95% CI: 73.1–81.1). Nearly half of the respondents reported feeling fatigued right now (55%), usual fatigue (57.7%) and worst fatigue (65.3%), as well as fatigue interference with daily activities, were higher(82.2%) (Fig 4).

## Factor associated with fatigue among adult cancer patients

Marital status, education level, occupation, BMI, social support, medical expense coverage, anemia, use of antiemetic, pain treatment modalities, type of cancer, stage of cancer, cancer treatment type, cycle of treatment, tobacco use, physical activity, anxiety, sleep disturbance, depression, and the number of comorbidities were among the 26 variables that were included in the bi-variable logistic regression and that were found to have p-values <0.25. Poor social support, patients with anemia, physically inactive, patients with anxiety, pain treatment modalities for cancer (surgery and surgery with chemotherapy), 6–7 cycle of treatment, and patients with underweight were significantly associated factors with CRF after accounting for potential confounding variables in the multivariable logistic regression.

Patients with poor social support were four times more likely to experience fatigue than those with strong social support (AOR = 3.62; 95% CI: 1.53–8.60). Cancer patients who were anemic had a twofold increased risk of fatigue compared to cancer patients who were not anemic (AOR = 2.01; 95% CI: 1.04–3.90). Clinically meaningful fatigue was nearly four times more likely to emerge in physically inactive patients than in physically active cancer patients (AOR = 3.63; 95% CI: 1.74–7.54). Patients with anxiety were three times more likely than cancer patients without anxiety to experience clinically significant fatigue (AOR = 3.13; 95% CI:

**Table 4. Psychosocial and behavioral factor among adult cancer patients receiving cancer treatment at oncology unit in Amhara region, Ethiopia, 2022 (N = 449).**

| Variable | Description | Frequency | 100% |
|---|---|---|---|
| Alcohol drinking | Low risk | 336 | 74.8% |
| | Moderate risk | 85 | 18.9% |
| | High risk | 28 | 6.2% |
| Risk level of tobacco smoking | Low risk | 399 | 88.9% |
| | Moderate risk | 6 | 1.3% |
| | High risk | 44 | 9.8% |
| Sleep disturbance | No sleep disturbance | 176 | 39.2% |
| | Mild sleep disturbance | 105 | 23.4% |
| | Moderate sleep disturbance | 111 | 24.7% |
| | Sever sleep disturbance | 57 | 12.7% |
| Social support | Poor social support | 213 | 47.4% |
| | Moderate social support | 160 | 35.6% |
| | Strong social support | 76 | 16.9% |

1.54–6.36). Similarly, patients who underwent surgery as well as chemotherapy had a nearly four-fold increased risk of experiencing fatigue compared to those who underwent only chemotherapy (AOR = 3.56; 95% CI: 1.68–7.54). However, cancer patients treated with surgery alone as opposed to chemotherapy alone had a 79% lower chance of experiencing fatigue (AOR = 0.21; 95% CI: 0.06–0.78). Patients who received treatment for fewer than three cycles

**Table 5. Clinical characteristics of adult cancer patients receiving cancer treatment at oncology unit in Amhara region, Ethiopia, 2022 (N = 449).**

| Variable | Description | Frequency | 100% |
|---|---|---|---|
| No of comorbidities | No comorbidities | 296 | 65.9% |
| | 1–2 | 95 | 21.2% |
| | 3–4 | 34 | 7.65 |
| | ≥5 | 24 | 5.3% |
| Nutritional status | Underweight | 233 | 51.9% |
| | Normal | 71 | 38.5% |
| | Overweight | 81 | 9.6% |
| Stage of cancer | Early stage | 104 | 23.2% |
| | Advanced stage | 272 | 60.6% |
| | Unknown stage | 73 | 16.3% |
| Cycle of treatments | <3 cycle | 153 | 34.1% |
| | 3–5 cycle | 199 | 44.3% |
| | 6–7 cycle | 52 | 11.6% |
| | >7 cycle | 45 | 10.0% |
| Cancer Treatment modalities | Chemotherapy | 217 | 48.3% |
| | Surgery | 24 | 5.3% |
| | Chemotherapy with surgery | 181 | 40.8% |
| | Chemotherapy with radiotherapy | 27 | 6% |
| Pain intensity level | Mild pain | 206 | 45.9% |
| | Moderate pain | 151 | 33.6% |
| | Sever pain | 92 | 20.5% |
| Pain treatment modalities | No treatment | 262 | 58.4% |
| | Non & Weak opioid | 173 | 15.8% |
| | Moderate & Strong opioid | 43 | 18.0% |

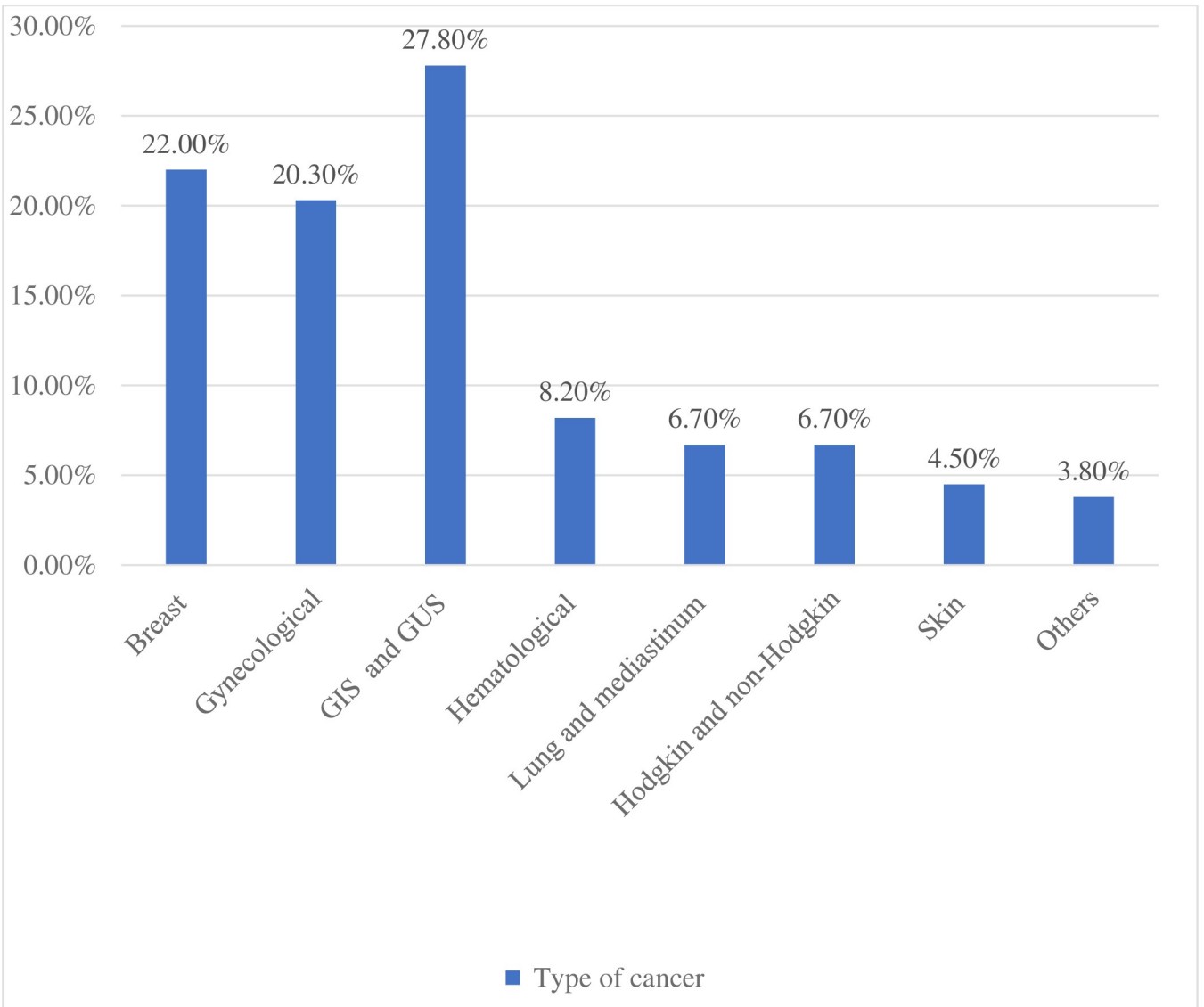

**Fig 3. Type of cancer among adult cancer patients receiving cancer treatment at oncology unit in Amhara region, Ethiopia, 2022 (N = 449).**

experienced a four-fold increased risk of fatigue compared to patients who received treatment for seven or more cycles (AOR = 4.43; 95% CI: 1.53–12.80). Additionally, compared to patients with normal nutritional status, adult cancer patients with undernutrition had a twofold higher risk of feeling fatigued (AOR = 2.03; 95% CI: 1.05–3.90) (Table 6).

## Discussion

The main purpose of this study was to assess the prevalence of fatigue and its associated factors among adult cancer patients receiving cancer treatment at the oncology unit in the Amhara region, Ethiopia. The overall prevalence of fatigue among adult cancer patients receiving cancer treatment was 77.3% at 95% CI: (73.1–81.1%). This study is in line with studies conducted in Iraq, 75% [11], Ireland 75% [48], South Africa 80% [49], and Addis Ababa Black Lion Hospital 74.8% [25]. This similarity is due to the fact that the pathological nature of cancer and

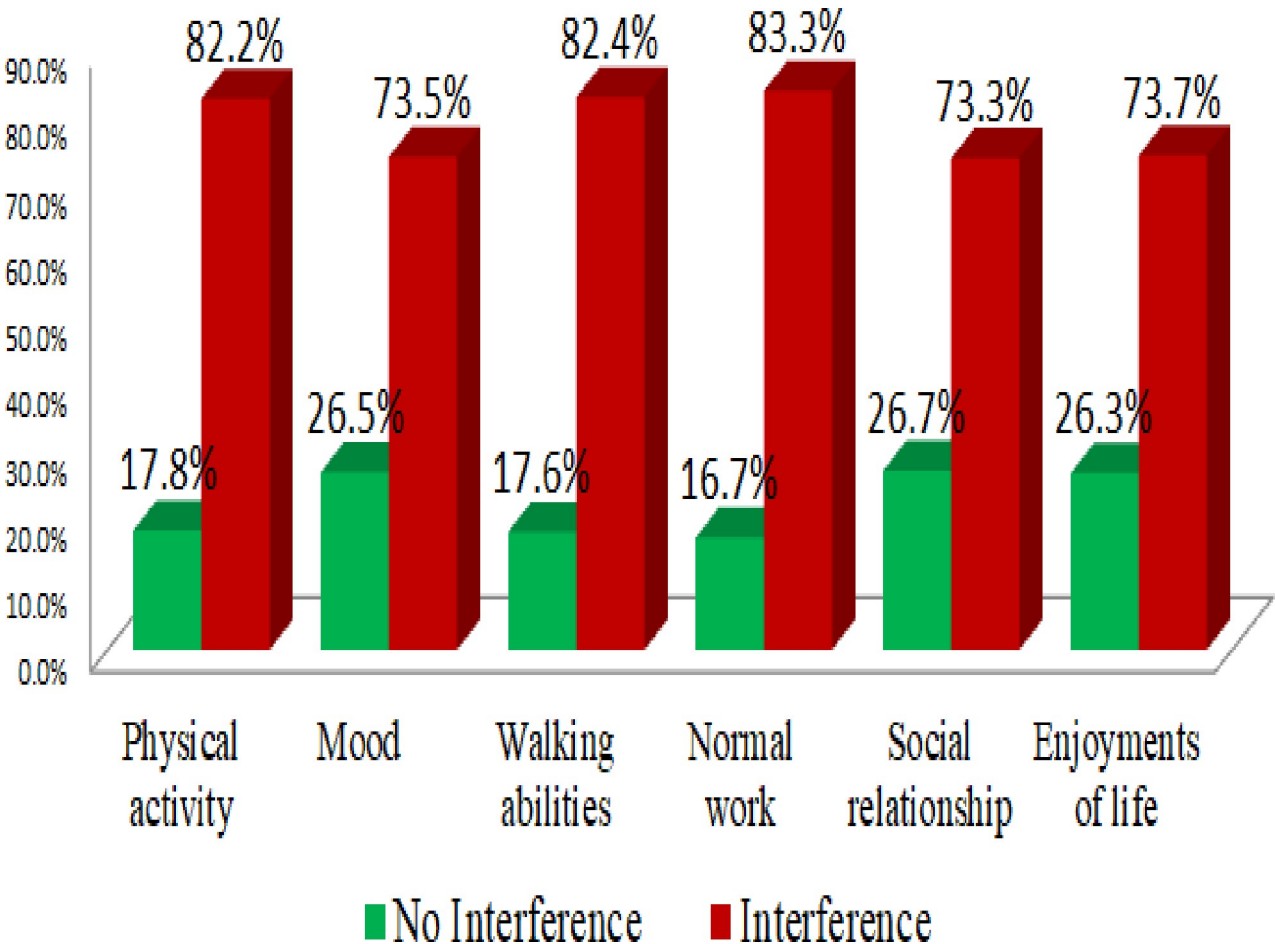

**Fig 4. Interference of fatigue on daily activities among adult cancer patients receiving cancer treatment at oncology unit in Amhara region, Ethiopia, 2022 (N = 449).**

treatments for cancer like chemotherapy, radiotherapy, and surgery can induce inflammation as a result of cellular damage and tissue injury to the regular structure and function of mitochondria [8]. Once the mitochondria are damaged, the energy supply is reduced, which results in fatigue [5,6,8].

However, the prevalence of the current study was much higher than studies which have been conducted in South Korea, 67.2% [50], Lebanon, 46.3% [51], Italy, 43.5%, [52], and Netherlands, 48% [53]. This discrepancy might be due to the difference in study population, sample size, tool, and eligibility criteria [6,54,55]. In the current study, all types of cancer were incorporated with different treatment options, whereas in Korea, the study was focused on only breast cancer treated by radiotherapy with a small sample size (n = 210). The cumulative effect of combination treatment modality also increases the risk of fatigue [56]. On the contrary, in Ethiopia there is low screening, health promotion and prevention programs [57] for cancer, and that is why cancer patients are visiting health institutions at an advanced stage (stage 3 and 4) (60.6%,) whereas in Korea, stage 1 and stage 2 accounted for 88.9%. In Lebanon only breast cancer, with the assessments of the European Organization for Research and Treatment of Cancer Quality of Life Questionnaire and a sample size of (n = 67), in Italy Functional Assessment of Cancer Therapy and a sample of (n = 133) and in the Netherlands, checklist of

**Table 6. Factors associated with fatigue among adult cancer patients receiving cancer treatment at oncology unit in Amhara region, Ethiopia, 2022 (N = 449).**

| Variable | Categories | Fatigue | | COR (95% CI) | AOR (95% CI) | P-Value |
|---|---|---|---|---|---|---|
| | | No | Yes | | | |
| Social support | Poor | 28 | 185 | 5.34(2.93–9.76) | 3.62(1.53–8.60) | **0.004** |
| | Moderate | 40 | 120 | 2.42 (1.36–4.32) | 1.78(0.80–3.96) | 0.16 |
| | Strong | 34 | 42 | 1 | 1 | |
| Anemia | No | 68 | 135 | 1 | 1 | |
| | Yes | 34 | 212 | 3.14(1.97–5.00) | 2.01(1.04–3.90) | **0.038** |
| Physical activity | No | 19 | 191 | 5.35(3.1–9.19) | 3.63(1.74–7.54) | **0.001** |
| | Yes | 83 | 156 | 1 | | |
| Anxiety | No | 67 | 105 | 1 | 1 | |
| | Yes | 35 | 242 | 0.23(0.14–0.36) | 3.13(1.54–6.36) | **0.002** |
| Pain treatment modalities | Chemotherapy | 57 | 160 | 1 | 1 | |
| | Surgery | 12 | 12 | 0.35(0.15–0.38) | 0.21(0.06–0.78) | **0.02** |
| | Chemotherapy & surgery | 21 | 160 | 2.71(1.57–4.68) | 3.56(1.68–7.54) | **0.001** |
| | Chemotherapy and radiotherapy | 12 | 15 | 0.44 (0.19–1.00) | 0.46(0.13–1.58) | 0.22 |
| Cycle of treatment | <3 cycle | 24 | 129 | 6.1(2.96–12.74) | 4.43(1.53–12.8) | 0.006 |
| | 3–5 cycle | 40 | 159 | 4.54(2.30–8.97) | 3.55(1.38–9.09) | 0.008 |
| | 6–7 cycle | 14 | 38 | 3.10(1.32–7.24) | 3.41(1.04–11.12) | **0.042** |
| | >7 cycle | 24 | 21 | 1 | 1 | |
| BMI | Normal | 55 | 118 | 1 | 1 | |
| | Underweight | 36 | 197 | 2.55 (1.58–4.11) | 2.03(1.05–3.90) | **0.04** |
| | Overweight | 11 | 32 | 1.35 (0.64–2.89) | 2.18(0.73–6.56) | 0.16 |

GUS = genitourinary system (prostate, renal, bladder), GIS = gastrointestinal system (colorectal the most common), stomach Ca, others: Include sarcoma, squamous cell Ca, axillary Ca, fibro ostium sarcoma, colangionic Ca, bone Ca, Gynecological Ca = cervical ca (the most common), ovarian ca, uterine ca, Ca = Cancer.

individual strength was used for assessment (n = 83), while in the present study, BFI was employed. Furthermore, it might be due to differences in healthcare provision across the country. The service for accessing quality of cancer treatment and palliative care in low-income countries, like Ethiopia, is poor and very challenging, which could exacerbate fatigue [57,58].

On the other hand, the present study was lower than studies conducted in Jordan, (87.5%) and India (83.3–88.18%) [59–61]. The discrepancy might be due to difference in study population [8,54], tool, sample size [60], and timing of assessment. The current study used the BFI scale to assess CRF and data was collected usually before receiving treatment at the outpatient department, whereas in Jordan, the Piper Fatigue Scale was used [59] and data were collected at the time of treatment. Previously studied information showed that fatigue is higher at the time of treatment and improves 7 to 10 days after treatment [62]. In India, only advanced cancer patients were studied using the Functional Assessment of Chronic Illness Therapy scale, whereas in the current study, all cancer patients were studied.

In this study, there were factors which had a significant association with CRF including poor social support, patients with anemia, physically inactive, patients with anxiety, pain treatment modalities for cancer (surgery and surgery with chemotherapy), 6–7 cycle of treatment, and patients with underweight.

Patients who had poor support from their families, neighbors, and friends was supported by studies conducted in Iran [63], and China [50]. This is due to that social support in terms of emotional (feeling love and having the certainty someone to trust), instrumental (availability of immediate help), and informational (receiving advice) are beneficial for improving physiological symptoms and therapeutic effects [64,65].

Patients who had anemia was another factor significantly associated with CRF. This finding was supported by studies conducted in India [60]. This could be due to cancer treatments, such as chemotherapy and radiotherapy, that target bone marrow and reduce blood count or red blood cell levels. When blood counts fall, the oxygen-carrying ability of the blood (hemoglobin) decreases, resulting in decreased tissue oxygenation and, eventually, tissue hypoxia [66], which causes a cessation of glucose metabolism at the lactate level and eventually leads to fatigue [67].

Physically inactive was the third factor associated with CRF and research from Australia [68], meta-analysis conducted in the USA [69], Netherland [70], China [32] and Switzerland [71] all corroborated this. This may be due to the fact that patients who engage in regular physical activity see improvements in their cardiorespiratory fitness, muscle function, and oxygen conservation [72], which improves their psychological well-being and physical fitness and reduces inflammation by raising levels of anti-inflammatory cytokines [73]. Additionally, exercise can lessen weariness indirectly by improving mood, immune system, or sleep [74].

Anxiety, which was supported by research from Australia [75], Netherland [76], and Nigeria [77] was the fourth factor in this study that had a significant association with CRF. This may be the result of biologically related processes; in particular, worry may lead to weariness via dysregulation of the hypothalamic-pituitary-adrenal axis [78]. As a result of internal and external stressors during therapy, the hypothalamic-pituitary-adrenal axis becomes dysregulated, depleting the body's supply of cortisol and impairing its ability to secrete it [79]. Pro-inflammatory cytokines are known to contribute to exhaustion, and they have been associated to low cortisol levels [6,80]. A range of physiological symptoms, such as an elevated heart rate, nausea, difficulty breathing, losing appetite, and sleep disruptions, as well as anxiety as subjective feelings of fear or worry brought on by cancer, may also result in fatigue [77].

Surgery and surgery combined with chemotherapy were the fifth important factor for CRF, which was corroborated by research from Addis Ababa Black Lion Hospital [25], as well as the Netherlands [56]. This is due to during surgery there is a dysfunctional energy metabolism, a reduction in adenosine triphosphate (ATP) synthesis due to cachexia or injury to the mitochondria, the loss of skeletal muscle, and the increased need for energy, which led to an increase in the onset of weariness [6,8].

In this study, the six significant factor that was associated with CRF was 6–7 cycle of treatment. The possible explanation might be due to the healing process of cancer being maximized as the patients fully cover their cycle of treatment and also the psychological factors like anxiety and depression being lowered and psychological adaptability being increased. However, this finding interestingly contrasts with a study conducted in Lebanon [51]. This might be due to a difference in measurement tool, inclusion, and sample size.

The final important factor associated to CRF was underweight. Studies in USA, Australia, and Jordan supports this claim [4,81–83]. This could be as a result of the pathologic nature of cancer and its treatment, which could result in nutritional issues like altered taste, anorexia, diarrhea, and unintended weight loss that could then weaken the cancer survivor and modify their body composition [81]. Additionally, cancer treatments may worsen already-present symptoms or have unfavorable side effects like taste loss, xerostomia, inadequate nutritional intake, nausea, and vomiting, which can cause involuntary muscular atrophy and weakness and increase fatigue due to altered ATP metabolism [78,82].

## Conclusion and recommendation

Nearly three fourth of cancer patients developed fatigue among adult cancer patients receiving cancer treatment at oncology units in Amhara region. POOR social support, having anemia,

being physically inactive, having anxiety, type of treatment modalities (combination of surgery and chemotherapy), cycle of treatment and undernutrition, were factors significantly associated with CRF.

Therefore, health care providers are better to screen fatigue early and create a social network with family, friends and psychiatrist (because family friends and psychiatrist are a significant source of support) for coping mechanism. In addition, better to advice patients to include variety of food type for consumption and to perform physical activity as tolerated, foster the linkage between cancer patients with nutritional problem and those who have physically inactive to nutritionist and physiotherapist and cheek their hemoglobin level for all cancer patients at each treatment period and manage accordingly. **Nurse** shall consider fatigue as one of the vital sign in cancer patients and incorporate in vital sign sheet format. Furthermore, education programs on fatigue, etiology and treatment shall be given for the patients.

### Researchers

Further qualitative and longitudinal studies are needed to explore the state of fatigue on cancer patients by incorporating the missed variables like drug and biomarker related variables.

### Limitation of the study

Some drug-related variables (e.g. chemotherapy medication) and certain biomarkers like electrolyte, and inflammatory biomarkers were not assessed.

### Supporting information

**S1 Data.**
(SAV)

### Acknowledgments

We are grateful to the participants from referral hospitals in Amhara region who provided us with the essential information. We'd also want to thank the data collectors and supervisors for their time and effort.

### Author Contributions

**Conceptualization:** Lingerew Animaw, Teshager Woldegiorgis Abate, Destaw Endeshaw, Dejen Tsegaye.

**Data curation:** Lingerew Animaw, Teshager Woldegiorgis Abate, Destaw Endeshaw.

**Formal analysis:** Lingerew Animaw, Teshager Woldegiorgis Abate, Destaw Endeshaw, Dejen Tsegaye.

**Funding acquisition:** Lingerew Animaw.

**Investigation:** Lingerew Animaw.

**Methodology:** Lingerew Animaw, Teshager Woldegiorgis Abate, Destaw Endeshaw, Dejen Tsegaye.

**Project administration:** Lingerew Animaw.

**Resources:** Lingerew Animaw, Teshager Woldegiorgis Abate.

**Software:** Lingerew Animaw, Teshager Woldegiorgis Abate, Destaw Endeshaw.

**Supervision:** Lingerew Animaw, Teshager Woldegiorgis Abate.

**Validation:** Lingerew Animaw, Teshager Woldegiorgis Abate, Destaw Endeshaw.

**Visualization:** Lingerew Animaw, Destaw Endeshaw.

**Writing – original draft:** Lingerew Animaw, Teshager Woldegiorgis Abate, Destaw Endeshaw.

**Writing – review & editing:** Lingerew Animaw, Teshager Woldegiorgis Abate, Destaw Endeshaw, Dejen Tsegaye.

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
