## [Decision Letter · Decision Letter 0]

22 Nov 2022

PONE-D-22-27844Fatigue And Associated Factors Among Adult Cancer Patients Receiving Cancer Treatment At Oncology Unit In Amhara Region, Ethiopia, 2022PLOS ONE

Dear Dr. Dejen Tsegaye,

Thank you for submitting your manuscript to PLOS ONE. After careful consideration, we feel that it has merit but does not fully meet PLOS ONE’s publication criteria as it currently stands. Therefore, we invite you to submit a revised version of the manuscript that addresses the points raised during the review process.

We look forward to receiving your revised manuscript.

Kind regards,

Muhammad Junaid Farrukh

Academic Editor

PLOS ONE

Journal Requirements:

https://www.mdpi.com/2073-4409/8/7/738/htm?

In your revision ensure you cite all your sources (including your own works), and quote or rephrase any duplicated text outside the methods section. Further consideration is dependent on these concerns being addressed.

Reviewers' comments:

Reviewer's Responses to Questions

**Comments to the Author**

1. Is the manuscript technically sound, and do the data support the conclusions?

Reviewer #1: Yes

2. Has the statistical analysis been performed appropriately and rigorously? 

Reviewer #1: Yes

3. Have the authors made all data underlying the findings in their manuscript fully available?

Reviewer #1: Yes

4. Is the manuscript presented in an intelligible fashion and written in standard English?

Reviewer #1: Yes

5. Review Comments to the Author

Reviewer #1: The topic is interesting discussing the prevalence of cancer related fatigue, a common symptom in cancer patients that is usually ignored and underestimated. Fatigue can significantly affect the patients’ quality of life.

General

• The article requires language editing.

Title:

• Appropriate and descriptive but I would prefer that the year should be removed from the title.

Abstract:

• “Fatigue among cancer”: I think it should be fatigue among cancer patients. kindly revise

• Results: with non-response rate of 9: what is the number 9? The rate is usually not a crude number it should be related to denominator as 9% or 9/1000 patients .

• Line 3 in the results: “physical inactive “ should be replaced with physical inactivity

• Kindly remove redundancy. The conclusion includes repetition of the results.

Introduction:

• Well written and descriptive.

Material and methods:

• Line 67: I think study area should be replaced with setting

• Population: the authors should simply mention that the eligible criteria is cancer patients receiving treatment. kindly remove redundancy.

• Line 84: “and” should be added before hearing or intellectually impaired

• Table 1 is not cited in text.

• It is not clear why the sample size was multiplied by 1.5.

• Sapling procedure is not clear. why stratified sampling was used ? what is the factor used for stratification?

• Operational definition: it would be better if summarized in a table

• How were all these questionnaires filled? Was that by face-to-face interview? How long did it take from each patient to film all these questions?

Results:

• Line 313: on why p-values was <0.25. selected to select variable?

• Line 334: it should be table 4 and should be cited in text.

• All data mentioned from line 319 to 333 are repeated in the table. Please remove unnecessary repetitions.

• Figures are not cited in text.

• Figures numbering is not correct

Discussion:

• Well written.

• The authors did not mention the limitations of their study.

6. PLOS authors have the option to publish the peer review history of their article (what does this mean?). If published, this will include your full peer review and any attached files.

Reviewer #1: No

---

## [Author Response · Author response to Decision Letter 0]

25 Nov 2022

Author’s Point-by-Point Response to the Reviewer's and Editors Reports

Fatigue And Associated Factors Among Adult Cancer Patients Receiving Cancer Treatment At Oncology Unit In Amhara Region, Ethiopia 

Corresponding Authors Dejen Tsegaye/ dejenetsegaye@gmail.com

Point by point response to Reviewers and Editors 

First and foremost, the authors would like to express their gratitude to the PLOSE ONE Journal editors and reviewers for thoroughly evaluating this work and offering the required corrections. We made changes based on the feedback we received and presented each comment point by point. The authors attempted to address all of the concerns expressed by the editorial board and reviewers. Please note that the response was written in blue font.

Authors' responses to the authors' remarks

REVIEWER #1

Comment: Title: Appropriate and descriptive but I would prefer that the year should be removed from the title.

Response: We appreciate your input, and we removed the year as a result.

Abstract: 

Comment: “Fatigue among cancer”: I think it should be fatigue among cancer patients. kindly revise

Response: We made the necessary corrections on the sentence in general. I'm grateful.

Comment: Results: with non-response rate of 9: what is the number 9? The rate is usually not a crude number it should be related to denominator as 9% or 9/1000 patients 

Response: The non-response rate, as shown in brackets, is 1.97%. The number "9" represents the number of respondents who did not respond. I appreciate your thoughtful perspective. We made the required correction.

Comment: Line 3 in the results: “physical inactive “ should be replaced with physical inactivity

Response: Thank you! We made the necessary change.

Comment: Kindly remove redundancy. The conclusion includes repetition of the results.

Response: Thank you for your suggestion. We removed the repeated part.

Introduction: Well written and descriptive. 

Response: Thank you!

Material and methods:

Comment: Line 67: I think study area should be replaced with setting

Response: Thank you, we replaced area with setting. Setting is the appropriate word.

Comment: Population: the authors should simply mention that the eligible criteria is cancer patients receiving treatment. kindly remove redundancy.

Response: Thank you for your comment. We made the necessary change.

Comment: Line 84: “and” should be added before hearing or intellectually impaired

Response: Thank you for your suggestion. We made the necessary change.

Comment: Table 1 is not cited in text.

Response: Thank you for your suggestion. We made the necessary change.

Comment: It is not clear why the sample size was multiplied by 1.5.

Response: As stated in a sentence in line "129," design effect has to be used in this study due to the variety of study settings and population types. Therefore, we multiplied the calculated sample size by 1.5 in order to have a sufficient sample size. I appreciate your asking for further information on this.

Comment: Sapling procedure is not clear. why stratified sampling was used ? what is the factor used for stratification? 

Response: I appreciate you asking. Stratification was discovered to be necessary in this investigation due to variations in participants' provinces of residence.

Comment: Operational definition: it would be better if summarized in a table

Response: Thank you for your suggestion. We summarized the operational definition in a table.

Comment: How were all these questionnaires filled? Was that by face-to-face interview? How long did it take from each patient to film all these questions?

Response: Face-to-face interviews, which last 30 minutes for each patient, were used to complete the questionnaire. We explained in the ‘’ Data collection tools and personnel’’ part. Thank you for helping us to add clarification on this part.

Results:

Comment: Line 313: on why p-values was <0.25. selected to select variable?

Response: We chose 0.25 since there were a lot of variables included in the multivariable analysis with a p value of 0.25. We would have used 0.3 if the number of variables had been lower.

Comment: Line 334: it should be table 4 and should be cited in text.

Response: It is table 5, and we mentioned it correctly. Due to its automatic citation, the error was made when it was transformed to pdf. I appreciate your thoughtful feedback.

Comment: All data mentioned from line 319 to 333 are repeated in the table. Please remove unnecessary repetitions.

Response: I appreciate the comment. We decided to leave it as-is because the text portion interprets the table. We don't think all readers will be able to comprehend or interpret the table in the same way. We will talk more and take the text portion out. I hope you will include your final observation or recommendation in the reviewer's second comment. Again, I'm grateful.

Comment: Figures are not cited in text.

Response: There are four figures incorporated in this study and Figure 1, 2, 3 and 4 are cited in line 131, 156, 273 and 283 respectively. Of these, figure 2 was missed. Thank you for the thoughtful comment.

Comment: Figures numbering is not correct

Response: After we correct the citation of figures accordingly, numbering is also corrected. Thank you for the comment.

Discussion:

Comment: Well written.

Response: Thank you!

Comment: The authors did not mention the limitations of their study.

Response: We mentioned our limitation in the study. Thank you!

We appreciate all of the reviewers' and editors' helpful feedback, suggestions, and questions.

Thank you,

With kind regards!

---

## [Editor Report · Decision Letter 1]

12 Dec 2022

Fatigue And Associated Factors Among Adult Cancer Patients Receiving Cancer Treatment At Oncology Unit In Amhara Region, Ethiopia, 2022

PONE-D-22-27844R1

Dear Dejen Tsegaye

We’re pleased to inform you that your manuscript has been judged scientifically suitable for publication and will be formally accepted for publication once it meets all outstanding technical requirements.

Kind regards,

Muhammad Junaid Farrukh

Academic Editor

PLOS ONE
---

## [Editor Report · Acceptance letter]

27 Dec 2022

PONE-D-22-27844R1 

Fatigue And Associated Factors Among Adult Cancer Patients Receiving Cancer Treatment At Oncology Unit In Amhara Region, Ethiopia 

Dear Dr. Tsegaye:

I'm pleased to inform you that your manuscript has been deemed suitable for publication in PLOS ONE. Congratulations! Your manuscript is now with our production department. 

Kind regards, 

on behalf of

Dr. Muhammad Junaid Farrukh 

Academic Editor

PLOS ONE